# A brief questionnaire measure of multidimensional schizotypy predicts interview-rated symptoms and impairment

**Kathryn C. Kemp**[1‡], **Alyssa J. Bathery**[1‡], **Neus Barrantes-Vidal**[2,3,4], **Thomas R. Kwapil**[1,5]*

**1** Department of Psychology, University of Illinois at Urbana-Champaign, Champaign, IL, United States of America, **2** Departament de Psicologia Clínica i de la Salut, Universitat Autònoma de Barcelona, Barcelona, Spain, **3** Sant Pere Claver–Fundació Sanitària, Barcelona, Spain, **4** CIBERSAM, Instituto de Salud Carlos III, Barcelona, Spain, **5** Department of Psychology, University of North Carolina at Greensboro, Greensboro, NC, United States of America

‡ Joint first authors.
* trkwapil@illinois.edu

**Data Availability Statement:** Data are posted on Open Science Framework https://osf.io/rky2d/.

## Abstract

The present study employed structured diagnostic interviews to assess the construct validity of the brief version of the Multidimensional Schizotypy Scale (MSS-B), which was developed to assess positive, negative, and disorganized dimensions of schizotypy. It was hypothesized that the MSS-B subscales would be associated with differential patterns of symptoms and impairment, comparable to findings for the full-length MSS. A total of 177 young adults completed structured diagnostic interviews assessing symptoms and impairment. As hypothesized, MSS-B positive schizotypy was significantly associated with interview ratings of positive (psychotic-like) symptoms, as well as schizotypal and paranoid personality disorder traits. MSS-B negative schizotypy was associated with interview ratings of negative symptoms, as well as schizoid, paranoid, and schizotypal traits. Furthermore, negative schizotypy predicted Cluster A personality disorder diagnoses. MSS-B disorganized schizotypy was associated with interview ratings of disorganized symptoms. All three schizotypy dimensions were associated with impaired functioning. This was the first study to evaluate the validity of the MSS-B using interview measures, and the pattern of findings for each MSS-B subscale was closely comparable to the findings for the full-length MSS. Contrary to our hypothesis, cannabis use was largely unassociated with psychotic-like symptoms and did not moderate the expression of the schizotypy dimensions. The MSS-B has good psychometric properties, high concordance with the full-length MSS, and good construct validity. Thus, it appears to be a promising brief alternative to traditional schizotypy measures.

**Funding:** The authors received no specific funding for this work.

**Competing interests:** The authors have declared that no competing interests exist.

# Introduction

## Schizotypy and schizophrenia

Current models consider schizophrenia to be the extreme manifestation of a dynamic continuum of symptoms and impairment referred to as schizotypy. Schizotypy ranges from mild, subclinical manifestations to the psychosis prodrome to full-blown psychosis [1, 2]. Schizotypy is multidimensional with positive, negative, and disorganized symptom dimensions consistent with the dimensions described in schizophrenia (e.g., [1, 3–5]). The positive dimension of schizotypy involves odd beliefs, unusual perceptual experiences, and suspiciousness, including full-blown delusions and hallucinations. The negative schizotypy dimension is characterized by diminished affect, cognition, and interest in the world, including alogia, anergia, flattened affect, avolition, and anhedonia. The disorganized schizotypy dimension involves disruptions in cognition, communication, and behavior that, at the extreme, involve formal thought disorder and grossly disorganized behavior [1]. Schizotypy offers a promising construct for understanding the etiology and development of schizophrenia-spectrum psychopathology, and the multidimensional structure provides a useful framework for resolving the heterogeneity of such conditions.

Numerous self-report inventories of schizotypy have been developed to: a) identify individuals at heightened risk for developing schizophrenia-spectrum disorders; b) examine the subclinical expressions of these forms of psychopathology; and c) provide a relatively brief, inexpensive, and non-invasive method of assessment [see reviews by 6–9]. Furthermore, in many cases, brief forms of these measures have been developed. For example, the 72-item Schizotypal Personality Questionnaire (SPQ; [10]) was shortened to a 22-item version (SPQ-B; [11]), the 104-item Oxford-Liverpool Inventory of Feelings and Experiences (O-LIFE; [12]) was shortened to a 43-item version (O-LIFE-SV; [13]), and the 166-item Wisconsin Schizotypy Scales (WSS; e.g., [14]) was shortened to a 60-item version (WSS-B; [15]). These measures were created in part to address time constraint issues that arise in many studies, especially when integrating schizotypy measures with large batteries of laboratory and neuroscience measures. Thus, having brief versions of scales that generally maintain the content coverage, reliability, and validity of the original scales is an important asset. Brief measures can also be used for initial screening to identify schizotypic participants who may later be assessed with the full-length version of the measure. In clinical settings, a briefer form of longer, more time-consuming questionnaires can be used to assess individuals who have difficulty harnessing their attention for an extended period of time (which is not an uncommon characteristic in schizotypy and schizophrenia).

## The multidimensional schizotypy scale-brief

Despite the widespread use of extant schizotypy measures, many of them suffer from conceptual and psychometric limitations, such as factor structures that are inconsistent with current conceptual models and outdated/biased items. Therefore, Kwapil et al. [16] developed the Multidimensional Schizotypy Scale (MSS) to assess current multidimensional formulations of schizotypy. The scale contains 77 true-false items that assess positive, negative, and disorganized schizotypy and was designed to map onto current multidimensional models of schizotypy and address psychometric limitations of currently available measures. The scale was developed following best practices (see [17]), and items were selected based upon content validity, classical test theory, item response theory, and differential item functioning. The positive, negative, and disorganized schizotypy subscales have good psychometric properties (e.g., [16, 18]), and questionnaire (e.g., [19]), interview [20], and ambulatory assessment [21] studies support their construct validity.

Gross et al. [22] developed a brief version of the scale (MSS-B) that takes only about five minutes for healthy participants to complete. Consistent with the development of the MSS, the 38 items in the MSS-B were derived from the full-length MSS based on classical test theory, item response theory, differential item functioning, and content validity. Despite its shortened length, the scale shows solid psychometric properties (e.g., [18]), and questionnaire studies (e.g., [19, 23]) support the construct validity of the three schizotypy subscales. Furthermore, Kemp et al. [18] demonstrated that the analogous subscales of the MSS and MSS-B demonstrate high concordance across separate testings, indicating that the MSS and MSS-B subscales tap comparable constructs.

Although multiple questionnaire studies have supported the construct validity of the MSS-B positive, negative, and disorganized schizotypy subscales, structured diagnostic interviewing provides a gold-standard method for assessing schizophrenia-spectrum symptoms and impairment, and for assessing the validity of schizotypy questionnaires. Multiple studies have demonstrated the construct validity of extant psychometric assessments using structured interviews, such as the Wisconsin Schizotypy Scales (e.g., [24–28]). In addition, Kemp et al. [20] recently employed structured interviews to assess the validity of the full-length MSS subscales. As hypothesized, they found that the schizotypy dimensions were associated with differential patterns of symptoms and impairment. Specifically, they reported that the MSS positive schizotypy subscale was associated with interview ratings of positive symptoms and schizotypal and paranoid personality disorder traits. MSS negative schizotypy was associated with interview-rated negative symptoms and schizotypal and schizoid traits, as well as diagnoses of schizophrenia-spectrum personality disorders. MSS disorganized schizotypy was associated with interview-rated disorganized symptoms and attentional deficits. All three schizotypy dimensions were associated with impaired global functioning. However, Smith et al. [29] stressed that support for the validity of the original measure does not automatically confer to brief forms and that reduction in items may limit content coverage even if the original and brief forms are highly correlated. Therefore, the present investigation provides the first study to examine the validity of the MSS-B using structured interviews.

## Goals and hypotheses of the present study

The goal of the present study was to examine the associations of positive, negative, and disorganized schizotypy, as assessed by the MSS-B, with interview measures of symptoms and impairment in a non-clinically ascertained sample of young adults recruited by Kemp, Bathery, et al. [20]. The interview measures assessed participants' past and current psychopathology, demographics, medical history, substance use, academic achievement, social functioning, and overall quality of life and adjustment. The present study adds to the previous literature by expanding our understanding of schizotypy by assessing the validity of the three-factor structure and further validating the MSS-B.

Based on previous interview studies (e.g., [24–28]), we hypothesized that high scores on the MSS-B positive schizotypy subscale would be associated with elevated interview-based ratings of positive schizotypy and schizotypal and paranoid personality traits. High scores on the negative schizotypy subscale were hypothesized to be associated with interview ratings of negative symptoms and schizoid and schizotypal personality traits. Based on findings by Kemp et al. [20], high scores on the disorganized schizotypy subscale were predicted to be associated with interview ratings of thought disorder and disorganization, mood disorder symptoms, alcohol/substance use and impairment, as well as deficits in attention. It was also expected that all three subscales would be associated with impaired functioning. Given that we assessed a non-clinically ascertained sample of young adults, we did not expect that many participants would

meet criteria for schizophrenia-spectrum disorders. However, to the extent that we did have such participants in the sample, we expected that they would score highly on the MSS-B subscales. Given the widespread reports that cannabis use exacerbates psychotic-like experiences and risk for schizophrenia-spectrum disorders (e.g., [30, 31]), we also conducted secondary analyses examining the extent to which self-reported cannabis use moderated the prediction of symptoms and impairment by the three schizotypy dimensions. Finally, given that schizotypy is best conceptualized as a multidimensional construct, we expected that the individual MSS-B subscales would account for more variance than a total MSS-B score that combines scores on each of the three subscales.

## Methods

### Participants

The present study used the sample assessed by Kemp et al. [20] that included 177 University of Illinois Urbana-Champaign (UIUC) undergraduate students recruited through a course-credit subject pool. The only exclusion criterion for enrolling in the study was that participants were required to be at least 18 years of age. The goal of the recruitment procedure was to have continuous distributions of scores on the three schizotypy dimensions that included adequate representation of high scorers on each dimension. Two related recruitment procedures were employed. First, anyone in the participant pool was allowed to sign up (this provided us with participants with a broad range of scores on the three schizotypy dimensions). Secondly, participants who had elevated scores (>1.5 standard deviations above the mean) on the MSS-B positive, negative, or disorganized schizotypy subscales taken during a prescreening were invited to sign up. This oversampling or enrichment procedure was successfully used in prior studies (e.g., [27]) to provide adequate representation of participants with elevated scores on the schizotypy dimensions. Note that although this process allowed us to invite high scorers to participate in the interview study, researchers were not able to match the specific MSS-B prescreening scores with any of the participants that enrolled in the study.

As noted in Kemp et al. [20], a priori power analyses determined that the sample size was sufficient to detect small effect sizes with power of at least .80. Demographic characteristics of the sample were: $M_{age}$ = 18.9 years, SD = 1.1, range 18 to 22 years; 61% female; 37% Caucasian, 13% African American, 27% Asian/Pacific Islander, 20% Hispanic, 3% other. The study was approved by the University of Illinois at Urbana-Champaign Institutional Review Board and all participants provided informed consent. Participants received course credit for taking part in the study.

### Materials

Participants were assessed individually. At the start of the session, they were administered a brief demographic questionnaire followed by the full version of the MSS that contains 77 true/false items assessing positive, negative, and disorganized dimensions. MSS-B scores for each participant were derived from the administration of the full MSS. Note that, as expected, the MSS and MSS-B analogous subscales correlated highly (positive schizotypy subscale, $r$ = .95; negative schizotypy subscale, $r$ = .96; disorganized schizotypy subscale, $r$ = .98).

This study employed validated structured interview measures that assessed demographic characteristics, symptoms, and impairment. A modified version of the overview section of the Structured Clinical Interview for the Diagnostic and Statistical Manual of Mental Disorders Fifth Edition (SCID-5; [32]) assessed demographic information, history of medical and mental health treatment, and a general overview of psychosocial functioning. The mood disorder module of the SCID-5 was administered, along with a modified version of the SCID-5

substance use disorder module [33] that assessed alcohol and other drug use and abuse. The scoring modification provided diagnoses of substance use disorders for nine classes of substances, as well as quantitative ratings of substance use and related impairment. For the purposes of examining whether cannabis use moderated the expression of the schizotypy dimensions, we used the rating of heaviest lifetime cannabis use, which ranged from 0 (never) to 4 (daily use for at least one month).

In order to assess clinical and subclinical manifestations of positive and disorganized schizotypy, we administered sections of the Structured Interview for Prodromal Symptoms (SIPS; [34]). Interview items derived from the SIPS were quantitatively rated on a scale of 0 (the symptom is absent) to 6 (severe and psychotic). We employed the interview-based Negative Symptom Manual (NSM; [35]) to assess subclinical and clinical levels of negative schizotypy symptoms. The NSM assesses five subdomains of negative schizotypy: anhedonia, social withdrawal, avolition/anergia, affective flattening, and alogia. An additional NSM subscale, not included in the total score, assesses attentional deficits. Each subdomain is rated from 0 (absent) to 8 (severe). The NSM was used because it provides comprehensive coverage of negative symptoms and, unlike other interview measures of negative symptoms, it does not appear to be highly saturated with depressive and positive symptoms [24].

Schizotypal, schizoid, and paranoid personality (Cluster A) traits and disorders were assessed using the International Personality Disorder Examination (IPDE; [36]). The IPDE is a structured interview designed to assess DSM-5 personality disorder traits and diagnoses. Each criterion is rated as 0 = not present, 1 = present, but subthreshold, or 2 = meets diagnostic threshold. Ratings of the number of criteria fully met and the sum of each criteria score were computed for each personality disorder, providing diagnoses and dimensional ratings for each of the personality disorders.

## Procedures

The methods and procedures used in this study follow those reported in Kemp et al. [20]. Participants completed the questionnaires (approximately 15 minutes) and structured interview (1 to 2 hours). Interviews were audio recorded and were conducted by a graduate student or a postbaccalaureate researcher, under the supervision of a licensed psychologist. The interviewers scored the interviews immediately after the assessment using a detailed scoring manual and a structured score sheet. Note that 31% of the interviews were independently scored by both interviewers to assess inter-rater reliability. Scoring disagreements were ultimately resolved by consultation with the senior investigator.

To assess the construct validity of each of the MSS-B subscales, linear regression analyses were computed for each of the quantitative dependent variables (e.g., NSM total score), and binary logistic regressions were used for dichotomously scored variables (e.g., presence or absence of major depressive disorder). In each regression analysis, we simultaneously entered scores on the three MSS-B schizotypy subscales as predictors of the interview ratings. This allowed us to examine the effect of each schizotypy dimension over-and-above the effect of the other dimensions, and the interaction terms over-and-above the main effects. For linear regressions, the standardized regression coefficient (β), change in $R$-squared, and the effect size ($f^2$) were all computed for each schizotypy predictor in each regression, along with the bivariate correlations ($r$). Note that following Cohen [37], $f^2$ values of .02 are considered small effects, .15 are considered medium effects, and .35 are considered large effect sizes. Bivariate correlations of .10, .30, and .50 are considered small, medium, and large, respectively. Odds ratio and 95% confidence interval were reported for binary logistic regressions. Secondary analyses examined the extent to which cannabis use moderated the schizotypy dimensions'

prediction of symptoms and impairment (other than the drug and alcohol-related outcome measures). These analyses involved entering cannabis use rating at step 2 of the regression analyses and the three cannabis x schizotypy dimension interactions at step 3 of the analyses.

## Results

### Multidimensional schizotypy scale descriptive statistics

De-identified data for the study is available on Open Science Framework at: https://osf.io/rky2d/ Table 1 presents the mean, standard deviation, range, and coefficient alpha of each of the MSS-B subscales. MSS-B positive, negative, and disorganized schizotypy subscale scores were converted to standardized scores based upon norms from 11,765 participants collected by our research team. The percentage of participants who scored 1.5 standard deviations above the mean was comparable across the three subscales: 14% for the positive schizotypy subscale, 12% on the negative schizotypy subscale, and 16% on the disorganized schizotypy subscale. The comparability of the mean scores on the MSS-B subscales and the proportion of individuals who scored particularly high indicates that we were successful in recruiting a sample that scored across the full range of the MSS-B subscales. The subscales displayed good internal consistency reliability, comparable with prior studies (e.g., [23]). Table 1 includes the intercorrelations of the three MSS-B subscales, which were slightly lower than samples seen in prior studies (e.g., [23]) and suggested that multicollinearity did not markedly impact the regression analyses.

### Quantitative interview measures of psychopathology and functioning

The descriptive statistics for the continuous interview measures are reported in Kemp et al. [20]. The inter-rater reliability for the 55 double-rated participants was good to excellent for all ratings (intraclass correlations with two-way mixed effects for single measures with absolute agreement ranged from .85 to .98). Table 2 reports a series of linear regression analyses in which the MSS-B positive, negative, and disorganized schizotypy subscale scores were simultaneously entered as predictors of each quantitative dependent measure. Note that the variance inflation factor values were all less than 1.1, indicating that the regression results were not adversely influenced by multicollinearity. In line with our hypotheses, MSS-B positive, negative, and disorganized schizotypy subscales had distinct patterns of associations with interview measures of symptoms and impairment. As hypothesized, all three schizotypy dimensions were associated with impaired functioning. Positive schizotypy was associated with elevated interview ratings of positive symptoms over-and-above the other two dimensions (large effect). MSS-B positive schizotypy was also significantly associated with interview ratings of schizotypal personality disorder traits (medium effect) and with paranoid personality disorder traits and disorganized symptoms (small effects), as well as inversely with negative and schizoid symptoms (small effects). MSS-B negative schizotypy was associated with negative symptoms and schizoid personality disorder traits (large effects), as well as schizotypal traits (medium effect) and paranoid traits (small effect). Negative schizotypy was also associated

**Table 1. Descriptive statistics and intercorrelations of the multidimensional schizotypy scale-brief (*n* = 177).**

| MSS-B Subscale | Descriptive Statistics | | | | Correlations | |
|---|---|---|---|---|---|---|
| | Mean | SD | Range | Coefficient Alpha | Negative Schizotypy | Disorganized Schizotypy |
| Positive Schizotypy | 2.49 | 2.45 | 0–9 | .74 | .04 | .12 |
| Negative Schizotypy | 2.10 | 2.76 | 0–13 | .84 | | .08 |
| Disorganized Schizotypy | 2.45 | 3.21 | 0–12 | .89 | | |

**Table 2. Linear regressions examining prediction by the multidimensional schizotypy scale-brief subscales ($n$ = 177).**

| Criteria: | MSS-B Positive Schizotypy | | | | MSS-B Negative Schizotypy | | | | MSS-B Disorganized Schizotypy | | | |
|---|---|---|---|---|---|---|---|---|---|---|---|---|
| | $r$ | $\beta$ | $\Delta R^2$ | $f^2$ | $r$ | $\beta$ | $\Delta R^2$ | $f^2$ | $r$ | $\beta$ | $\Delta R^2$ | $f^2$ |
| Global Functioning | -.19** | -.131** | .017 | .03 | **-.49*** | **-.455*** | **.206** | **.35** | **-.43*** | **-.375*** | **.138** | **.23** |
| SIPS-P total | ***.54**** | ***.517**** | ***.263*** | ***.41*** | .28*** | .254*** | .064 | .10 | .17* | .082 | .007 | .01 |
| SIPS-D total | .24** | .167** | .027 | .05 | .22** | .173*** | .030 | .05 | ***.61**** | ***.571**** | ***.319*** | ***.55*** |
| Negative Symptoms | -.06 | -.117* | .013 | .03 | ***.63**** | ***.617**** | ***.378*** | ***.72*** | .29*** | .260*** | .066 | .13 |
| NSM Attention | .19* | .124 | .015 | .02 | .10 | .056 | .003 | .00 | ***.53**** | ***.514**** | ***.258*** | ***.37*** |
| Schizotypal Symptoms | ***.41**** | ***.375**** | **.138** | **.22** | .46*** | .437*** | **.190** | **.30** | .20** | .117 | .013 | .02 |
| Schizoid Symptoms | -.14 | -.164** | .027 | .04 | ***.61**** | ***.614**** | ***.375*** | ***.62*** | .04 | .015 | .000 | .00 |
| Paranoid Symptoms | .21** | .188** | .035 | .04 | .30*** | .289*** | .083 | .10 | .10 | .055 | .003 | .00 |
| Alcohol Use | .04 | .042 | .002 | .00 | -.20** | -.207** | .043 | .05 | .03 | .045 | .002 | .00 |
| Alcohol Impairment | .11 | .108 | .011 | .01 | -.13 | -.140 | .019 | .02 | .06 | .059 | .003 | .00 |
| Cannabis Use | .07 | .050 | .002 | .00 | -.11 | -.121 | .015 | .01 | .16* | .160* | .025 | .03 |
| Drug Use | .03 | .022 | .000 | .00 | .03 | .021 | .000 | .00 | .06 | .050 | .002 | .00 |
| Drug Impairment | .03 | .019 | .000 | .00 | -.14 | -.151* | .023 | .02 | .16* | .165* | .027 | .03 |

*$p < .05$

**$p < .01$

***$p < .001$

medium effect sizes ($f^2$) in bold, large effect sizes in bold and italics

Each row represents a separate linear regression analysis in which the three MSS-B subscales were entered simultaneously as predictors to examine their unique prediction of each of the quantitative interview measures

with positive and disorganized symptoms (small effects). MSS-B disorganized schizotypy had its strongest associations with SIPS disorganized symptoms and attentional deficits (large effects) and was associated with negative symptoms (small effect). Regarding substance use and impairment, disorganized schizotypy was associated with cannabis use and impairment related to drug use (small effects). Positive schizotypy was unassociated with all of the substance-related dependent measures, whereas negative schizotypy was inversely associated with alcohol use and drug impairment (small effects).

## Categorical interview measures of psychopathology and functioning

Descriptive statistics for the categorical interview measures are presented in Table 3. Each dependent measure was dichotomous and reflects the percentage of participants who endorsed the characteristic. Given that only four participants endorsed having no close friends and only seven participants reported one close friend, we dichotomized the 'close friends' score so that a score of 0 denotes having fewer than two close friends, and a score of 1 indicates having two or more friends. Inter-rater reliability was computed using the kappa statistic and was adequate to excellent for each of the categorical measures. Table 3 also presents the results of the binary logistic regression analyses. Note that none of the participants met criteria for a psychotic disorder; however, four participants qualified for diagnoses of schizoid personality and one participant met 3 of 4 criteria needed for the diagnosis. Of note, these five subjects were the 1st, 2nd, 4th, 5th, and 7th highest scorers on the MSS-B negative schizotypy subscale. Note that binary logistic regression of the MSS-B subscales predicting schizoid personality disorder could not be calculated because complete separation (perfect fit) occurred. One other participant, who was also elevated on the MSS-B negative schizotypy subscale, met criteria for schizotypal personality disorder. MSS-B negative schizotypy significantly predicted diagnoses of any Cluster A personality disorder.

**Table 3. Binary logistic regressions examining prediction by the multidimensional schizotypy scale-brief subscales ($n$ = 177).**

| Criteria: | % Endorsed | Kappa | MSS-B Positive Schizotypy | | MSS-B Negative Schizotypy | | MSS-B Disorganized Schizotypy | |
|---|---|---|---|---|---|---|---|---|
| | | | Odds Ratio | 95% CI | Odds Ratio | 95% CI | Odds Ratio | 95% CI |
| Never Dated | 26.0% | .95 | .98 | .70–1.36 | 1.34* | 1.02–1.77 | .86 | .62–1.18 |
| <2 Close Friends | 6.2% | 1.00 | .69 | .35–1.36 | 1.99*** | 1.33–2.97 | .97 | .57–1.67 |
| Mental Health Treatment | 36.2% | .84 | .99 | .72–1.34 | .92 | .69–1.23 | 1.76*** | 1.31–2.34 |
| Any Cluster-A PD | 2.8% | .85 | .54 | .16–1.78 | 4.17*** | 1.85–9.44 | .35 | .05–2.44 |
| Major Depressive Episode | 30.5% | .76 | 1.16 | .85–1.58 | 1.02 | .77–1.36 | 1.68*** | 1.27–2.24 |
| Manic/Hypomanic Episode | 5.6% | .85 | .98 | .52–1.84 | .94 | .53–1.69 | .87 | .46–1.64 |
| Suicidal Ideation | 20.3% | .74 | .84 | .57–1.23 | 1.36* | 1.01–1.83 | 1.81*** | 1.33–2.46 |
| Alcohol Use Disorder | 2.8% | .66 | .38 | .10–1.53 | .51 | .15–1.75 | 1.91* | 1.01–3.62 |
| Any Drug Use Disorder | 6.8% | 1.00 | 1.04 | .59–1.84 | .84 | .47–1.52 | 1.02 | .61–1.71 |
| Cannabis Use Disorder | 6.2% | 1.00 | 1.14 | .64–2.01 | .86 | .47–1.57 | 1.08 | .64–1.81 |

*$p < .05$

**$p < .01$

***$p < .001$

Each row represents a separate binary logistic regression analysis in which the three MSS-B subscales were entered simultaneously as predictors to examine their unique prediction of each of the categorical interview measures

As hypothesized, the negative schizotypy dimension was uniquely associated with endorsement of never having been in a serious or long-term dating relationship and with having fewer than two close friends. Interestingly, negative schizotypy was also significantly associated with a history of suicidal ideation. The MSS-B disorganized schizotypy subscale was associated with a history of mental health treatment, diagnosis of a past or current depressive disorder, history of suicidal ideation, and history of an alcohol use disorder.

## Does cannabis use moderate the expression of multidimensional schizotypy?

Over half of our sample (53%) reported using cannabis at some point in their lifetime, and 40% of the sample reported using within the last month. Among participants who reported using cannabis, approximately half reported single use or only occasional experimentation, whereas the other half reported using at least twice weekly for at least one month. S1 Table presents the bivariate associations of the rating of heaviest lifetime cannabis use with quantitative and categorical outcome measures (excluding substance-related dependent variables). S2 Table and S3 Table present the cannabis x schizotypy interaction terms in the prediction of quantitative and categorical outcome measures, respectively. Cannabis use was generally unassociated with the outcome measures (both at the zero-order and after partialling out the MSS-B subscales). However, not surprisingly, cannabis use did have significant bivariate associations with impaired functioning and attentional deficits. Furthermore, only one of the 45 cannabis x schizotypy interaction terms was significant. Thus, cannabis use did not moderate the expression of schizotypy in the present sample.

## Comparing the MSS-B subscales and total score

Given that schizotypy is conceptualized as a multidimensional construct, it is recommended that researchers use the positive, negative, and disorganized schizotypy subscales, and not employ a total MSS-B schizotypy score [16, 22]. The multidimensional factor structure of

**Table 4. Variance accounted for by MSS-B subscales and MSS-B total score (*n* = 177).**

| Criteria | Variance accounted by MSS-B Total Score | Variance accounted for by MSS-B Subscales | % of Subscale variance accounted by MSS-B Total |
|---|---|---|---|
| Global Functioning | 0.368 | 0.408 | 90.1% |
| SIPS-Positive Symptoms | 0.247 | 0.362 | 68.2% |
| SIPS-Disorganized Symptoms | 0.359 | 0.424 | 84.7% |
| NSM Negative Symptoms | 0.232 | 0.474 | 48.9% |
| NSM Attentional Deficits | 0.217 | 0.303 | 71.6% |
| Schizotypal Symptoms | 0.303 | 0.376 | 80.6% |
| Schizoid Symptoms | 0.079 | 0.398 | 19.8% |
| Paranoid Symptoms | 0.099 | 0.131 | 75.6% |
| Alcohol Use | 0.004 | 0.045 | 8.8% |
| Alcohol Impairment | 0.000 | 0.034 | 1.2% |
| Cannabis Use | 0.005 | 0.041 | 12.2% |
| Drug Use | 0.004 | 0.004 | 92.3% |
| Drug Impairment | 0.001 | 0.047 | 2.1% |

The final column shows how much variance the MSS-B total score accounts for relative to the total variance accounted for by the three MSS-B subscales.

schizotypy is demonstrated by the fact that positive, negative, and disorganized schizotypy show differential patterns of associations with interview ratings of symptoms and impairments. In order to examine whether a total score accounts for a comparable amount of information relative to the individual subscales, we computed an MSS-B total score and correlated it with our interview outcome measures listed in Table 2. The final column in the table shows how much variance the MSS-B total score accounts for relative to the total variance accounted for by the three MSS-B subscales. In every case, the variance accounted for by the MSS-B total score was less than the total variance accounted for by the three subscales (even though the same items were included in both analyses), thus indicating that using a total schizotypy score produces a marked loss of information (see Table 4). On average, the MSS-B total score only accounted for 50% of the variance accounted for by the subscales.

## Discussion

Nearly all current developmental psychopathology models of schizophrenia-spectrum psychopathology recognize that schizophrenia and related disorders represent the most severe manifestations of a broader continuum of subclinical and clinical symptoms and impairment. Such symptoms and impairment are seen in psychotic patients often long before their initial psychotic episode (as well as during residual phases) and in the nondisordered relatives of patients. These models incorporate phenotypes such as the psychosis prodrome [38], at-risk mental states [39], and ultra-high-risk status [40]. Furthermore, this continuum includes milder forms of schizophrenic-like symptoms and impairment that convey elevated risk for developing schizophrenia-spectrum disorders (e.g., [25, 41]). Schizotypy offers a unifying framework that incorporates all of these related conditions. The multidimensional structure of schizotypy offers a powerful approach for parsing the heterogeneity of schizophrenia-spectrum psychopathology. Finally, it provides a useful platform for studying risk, resilience, and the development and trajectories of these symptoms and disorders.

Psychometric assessment offers a powerful approach for assessing schizotypy that can easily be integrated with traditional laboratory, neuroscience, family, and ambulatory assessments

(e.g., [7]). Schizotypy questionnaires are relatively inexpensive and noninvasive, and can be administered to large numbers of participants at once, especially using online assessment methods. The full-length MSS offers a promising measure that maps onto current multidimensional models of schizotypy. The MSS was developed to overcome many of the limitations of extant schizotypy measures. It offers good internal consistency and test-retest reliability [16, 18], and interview [20], questionnaire [19], and ambulatory assessment [21] studies have demonstrated the validity of the schizotypy subscales.

Although full-length schizotypy scales such as the MSS, WSS, OLIFE, and SPQ have been widely used, investigators at times find their length to be problematic, which may limit their inclusion especially in studies that include extensive assessment batteries. Therefore, shortened forms can provide promising alternatives, especially as initial screening measures. However, shortened forms can suffer from diminished reliability and content coverage compared to their full-length versions, as well as loss of subscale inclusion. The MSS-B items were selected from the full-length MSS and represent "the best of the best" items in the original scale. The MSS-B subscales show little shrinkage in terms of coefficient alpha reliability (consistent with reductions predicted by the Spearman-Brown formula; [22]). However, as noted by Smith et al. [29], the fact that a short scale maintains good psychometric properties and associations with the original scale does not guarantee the validity of the brief version. They advocate that validity has to be demonstrated for the shortened versions.

The present study was the first investigation to employ structured interviews to validate the MSS-B. Consistent with previous interview and questionnaire studies using the MSS (e.g., [20]), and previous questionnaire studies using the MSS-B (e.g., [23]), the present study found that the positive, negative, and disorganized schizotypy subscales were associated with hypothesized, differential patterns of symptoms and impairment. These findings are especially notable given that the MSS-B subscales predicted symptoms and impairment in a non-clinically ascertained sample of young adults. Several findings are worth noting. First, all three schizotypy dimensions were uniquely associated with impaired functioning as assessed by the GAF (i.e., each MSS-B subscale was associated with impairment after partialling out the other two subscales). As expected, positive schizotypy predicted psychotic-like experiences (large effect size), schizotypal personality disorder traits (medium effect), and paranoid personality disorder traits (small effect). The results for negative schizotypy were especially striking as the subscale predicted diagnoses of Cluster A personality disorders (driven largely by the presence of schizoid personality disorder). MSS-B negative schizotypy had its strongest associations with negative symptoms and schizoid personality disorders traits (large effects), as well as schizotypal personality traits (medium effect). Finally, MSS-B disorganized schizotypy had its strongest associations with interview ratings of SIPS disorganized symptoms and NSM attentional deficits (large effects). Consistent with a number of previous studies indicating that disorganized schizotypy is associated with depression and negative affect (e.g., [19, 21]), disorganized schizotypy was also associated with history of depressive disorders, suicidal ideation, and mental health treatment.

Although a number of extant schizotypy measures advocate for the use of total schizotypy scores, we strongly recommend the separate use of the three MSS and MSS-B subscales. The present findings, as well as previous studies, demonstrate that the positive, negative, and disorganized subscales have unique patterns of associations with psychopathology measures. Furthermore, the use of a total schizotypy score only accounts, on average, for about one-half of the variance in measures of symptoms and impairment. Similar reductions in variance were reported in interview [20], questionnaire [42], and cognitive [43] studies.

The present study provides additional support for the use of the MSS-B as a brief, non-invasive measure of positive, negative, and disorganized schizotypy. Although the full-length MSS

provides slightly better reliability than the MSS-B, the MSS-B appears to offer a useful alternative. We suggest that researchers may find the MSS-B useful as a screening measure, for example in large testing pools. The MSS-B subscales have good test-retest reliability and good correspondence with the analogous MSS subscales across 3- to 7-week intervals [18]. The MSS-B may also be especially useful in large field studies or neuroscience studies, which often have extensive testing protocols and limited time for additional measures. Although we offer the MSS-B as an alternative to the MSS, its psychometric properties and preliminary validity findings suggest it can stand on its own as a first-choice measure.

An extensive literature suggests that cannabis use is associated with the development of psychotic-like symptoms [44] and psychotic disorders [45]. However, contrary to our expectations, our secondary analyses generally found that cannabis use was largely unassociated with schizotypic psychopathology, and it did not moderate the association of the MSS-B schizotypy subscales with schizotypic symptoms. These null findings do not seem to represent a lack of cannabis use by our sample, as approximately one-half of the sample reported using cannabis at some point in their life and approximately one-quarter of the sample reported regular use. It may be that the effects of cannabis in a relatively high functioning sample are more modest and may present across a longer time period.

The present study must be interpreted in light of several limitations. First, MSS-B positive, negative, and disorganized schizotypy scorers were derived from the administration of the full-length MSS, not administration of the MSS-B. However, it should be noted that a number of studies have compared the psychometric properties [18] and validity [23] of the MSS-B and found comparable results regardless of whether the MSS-B was administered or the MSS-B scores were derived from the MSS. The present study also employed a non-clinically ascertained sample of college students. Concerns are often raised about the use of college student samples for studying subclinical expressions of psychopathology and risk for clinical disorders. However, we believe that many of these concerns are misstated or overstated. As noted by Kemp et al. [20], college students are at an ideal age for assessing schizotypy, as they are just entering the window of risk for developing schizophrenia-spectrum symptoms and disorders. Furthermore, as demonstrated in this and other schizotypy interview studies (e.g., [26, 46]), college students readily experience schizophrenia-spectrum psychopathology (as well as other forms of psychopathology). Specifically in the present study, the MSS-B subscales predicted a wide variety of symptoms and impairment, including Cluster A personality disorders, despite the fact that the study employed a young, high functioning sample. Needless to say, future validation studies should extend these findings to assess more diverse samples, including community and clinically identified participants.

In summary, the present findings provide further support for the construct validity of the MSS-B as a measure of multidimensional schizotypy. The pattern of findings and effect sizes was nearly identical with the findings from Kemp et al. [20] for the full-length MSS. Furthermore, the findings lend additional support for the utility of the multidimensional model of schizotypy as a powerful framework for studying the etiology, expression, and development of schizophrenia-spectrum psychopathology. Finally, the study supports the use of psychometric assessment as a tool for assessing schizotypy and for screening participants to enhance the power of laboratory, clinical, and neuroscience assessments.

## Supporting information

**S1 Table. Bivariate correlations of cannabis use and interview measures of symptoms and impairment.**
(DOCX)

**S2 Table. Linear regressions examining prediction by the multidimensional schizotypy scale-brief subscales and cannabis use (n = 177).**
(DOCX)

**S3 Table. Binary logistic regressions examining prediction by the multidimensional schizotypy scale-brief subscales and cannabis use ($n$ = 177).**
(DOCX)

## Author Contributions

**Conceptualization:** Kathryn C. Kemp, Alyssa J. Bathery, Thomas R. Kwapil.

**Data curation:** Kathryn C. Kemp, Alyssa J. Bathery.

**Formal analysis:** Alyssa J. Bathery, Thomas R. Kwapil.

**Investigation:** Kathryn C. Kemp, Alyssa J. Bathery, Thomas R. Kwapil.

**Methodology:** Kathryn C. Kemp, Alyssa J. Bathery, Thomas R. Kwapil.

**Project administration:** Thomas R. Kwapil.

**Supervision:** Neus Barrantes-Vidal, Thomas R. Kwapil.

**Writing – original draft:** Kathryn C. Kemp, Alyssa J. Bathery, Thomas R. Kwapil.

**Writing – review & editing:** Kathryn C. Kemp, Neus Barrantes-Vidal, Thomas R. Kwapil.

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
