## [Decision Letter · Decision Letter 0]

22 Jul 2020

PONE-D-20-11293

A Brief Questionnaire Measure of Multidimensional Schizotypy Predicts Interview-rated Symptoms and Impairment

PLOS ONE

Dear Dr. Kwapil,

Thank you for submitting your manuscript to PLOS ONE. After careful consideration, we feel that it has merit but does not fully meet PLOS ONE’s publication criteria as it currently stands. Therefore, we invite you to submit a revised version of the manuscript that addresses the points raised during the review process.

We look forward to receiving your revised manuscript.

Kind regards,

Sinan Guloksuz, M.D., Ph.D.

Academic Editor

PLOS ONE

Journal Requirements:

2.We note that you have stated that you will provide repository information for your data at acceptance. Should your manuscript be accepted for publication, we will hold it until you provide the relevant accession numbers or DOIs necessary to access your data. If you wish to make changes to your Data Availability statement, please describe these changes in your cover letter and we will update your Data Availability statement to reflect the information you provide.

Reviewers' comments:

Reviewer's Responses to Questions

**Comments to the Author**

1. Is the manuscript technically sound, and do the data support the conclusions?

Reviewer #1: Yes

2. Has the statistical analysis been performed appropriately and rigorously? 

Reviewer #1: Yes

3. Have the authors made all data underlying the findings in their manuscript fully available?

Reviewer #1: Yes

4. Is the manuscript presented in an intelligible fashion and written in standard English?

Reviewer #1: Yes

5. Review Comments to the Author

Reviewer #1: The authors describe a study in which they evaluate the correspondence between a new, brief measure of multidimensional schizotypy (MSS-B), interview-based ratings of schizotypy, and other related outcomes. They find the expected associations between dimensions of schizotypy as assessed with the MSS-B and the interview-based measures.

This an extremely clear, well-written, and straightforward paper that lends further support to a new, brief tool for assessing dimensions of schizotypy. I think it will make a nice contribution to the schizotypy literature. I have minimal comments for the authors.

* Since the aim, hypotheses, methods, and results are so straightforward, I thought parts of the paper (e.g. the discussion) could be shorter.

* I’m curious as to why the authors assessed negative symptoms with the NSM as opposed to the negative symptom scores from the SIPS.

* Could the authors comment on how the associations in the current manuscript compare to those with the full MSS described in Kemp et al.?

* I’m not sure I understand the third column of table 4 - how is that value calculated?

6. PLOS authors have the option to publish the peer review history of their article (what does this mean?). If published, this will include your full peer review and any attached files.

Reviewer #1: No

---

## [Author Response · Author response to Decision Letter 0]

27 Jul 2020

We attached a full response to the Editor's and Reviewer's comment in our attached "response to reviewers" document.

---

## [Editor Report · Decision Letter 1]

30 Jul 2020

A Brief Questionnaire Measure of Multidimensional Schizotypy Predicts Interview-rated Symptoms and Impairment

PONE-D-20-11293R1

Dear Dr. Kwapil,

We’re pleased to inform you that your manuscript has been judged scientifically suitable for publication and will be formally accepted for publication once it meets all outstanding technical requirements.

Kind regards,

Sinan Guloksuz, M.D., Ph.D.

Academic Editor

PLOS ONE
---

## [Editor Report · Acceptance letter]

31 Jul 2020

PONE-D-20-11293R1 

A Brief Questionnaire Measure of Multidimensional Schizotypy Predicts Interview-rated Symptoms and Impairment 

Dear Dr. Kwapil:

I'm pleased to inform you that your manuscript has been deemed suitable for publication in PLOS ONE. Congratulations! Your manuscript is now with our production department. 

Kind regards, 

on behalf of

Dr. Sinan Guloksuz 

Academic Editor

PLOS ONE